# Epidemiology of scrub typhus and other rickettsial infections (2018–22) in the hyper-endemic setting of Mizoram, North-East India

**Vanramliana**[1ʘ], **Lalfakzuala Pautu**[1,2ʘ]*, **Pachuau Lalmalsawma**[2], **Gabriel Rosangkima**[1], **Devojit Kumar Sarma**[3], **Hunropuia Chinzah**[1], **Yogesh Malvi**[2], **Naveen Kumar Kodali**[4], **Christiana Amarthaluri**[5], **Karuppusamy Balasubramani**[6], **Praveen Balabaskaran Nina**[4,5]*

**1** Department of Life Sciences, Pachhunga University College, Mizoram University, Mizoram, India, **2** Integrated Disease Surveillance Programme, Health & Family Welfare Department, Mizoram, India, **3** ICMR- National Institute for Research in Environmental Health, Bhopal Bypass Road, Bhauri, Bhopal, Madhya Pradesh, India, **4** Department of Epidemiology and Public Health, Central University of Tamil Nadu, Thiruvarur, Tamil Nadu, India, **5** Department of Public Health and Community Medicine, Central University of Kerala, Kasaragod, Kerala, India, **6** Department of Geography, Central University of Tamil Nadu, Thiruvarur, Tamil Nadu, India

ʘ These authors contributed equally to this work.
* mafaka@pucollege.edu.in (LP); praveen.nina@cukerala.ac.in (PBN)

**Data Availability Statement:** Data is available in the supplementary material.

## Abstract

### Background

In the past decade, scrub typhus cases have been reported across India, even in regions that had no previous history of the disease. In the North-East Indian state of Mizoram, scrub typhus cases were first recorded only in 2012. However, in the last five years, the state has seen a substantial increase in the scrub typhus and other rickettsial infections. As part of the public health response, the Mizoram Government has integrated screening and line listing of scrub typhus and other rickettsial infections across all its health settings, a first in India. Here we detail the epidemiology of scrub typhus and other rickettsial infections from 2018–2022, systematically recorded across the state of Mizoram.

### Methodology/principal findings

The line-listed data positive for scrub typhus and other rickettsial infections identified by rapid immunochromatographic test and/or Weil-Felix test from 2018–22 was used for the analysis. During this period, 22,914 cases of rickettsial infections were recorded, out of which 19,651 were scrub typhus cases. Aizawl is the worst affected, with 10,580 cases (46.17%). The average incidence of rickettsial infections is 3.54 cases per 1000 persons-year, and the case fatality rate is 0.35. Only ∼2% of the reported scrub typhus cases had eschar. Multivariate logistic regression analysis indicate patients with eschar (aOR = 2.5, p<0.05), occupational workers [farmers (aOR:3.9), businessmen (aOR:1.8), construction workers (aOR:17.9); p<0.05], and children (≤10 years) (aOR = 5.4, p<0.05) have higher odds of death due to rickettsial infections.

**Funding:** This work was supported by Department of Health Research, DHR (R.11013/11/2021-GIA/ HR) [K.B.] and Science and Engineering Research Board, New Delhi, under Core Research Grant (CRG/2019/003016/BHS; dated 7th February 2022) [V.]. The funders had no role in study design, data collection and analysis, decision to publish, or preparation of the manuscript.

**Competing interests:** The authors have declared that no competing interests exist.

## Conclusion

The integration of systematic surveillance and recording of rickettsial diseases across Mizoram has shed important insights into their prevalence, morbidity, and mortality. This study underscores the importance of active surveillance of rickettsial infections across India, as the burden could be substantially higher, and is probably going undetected.

## Author summary

Even though scrub typhus and other rickettsial infections are continuously being recorded across India, there has been no systematic surveillance and reporting of the disease in any state. Mizoram, the North-East Indian state sandwiched between Bangladesh and Myanmar has integrated screening and systematic line listing of rickettsial infections across all its health centers. An important outcome of this effort is the very low case fatality rate (CFR); at 0.35, the CFR of rickettsial infections in Mizoram is very low when compared to the median CFR of ∼6% reported from other locations. These findings could serve as a template that could be replicated in other regions across India to prevent and control this neglected disease of huge public health importance.

## Introduction

Scrub typhus is a re-emerging infectious zoonotic disease caused by *Orientia tsutsugamushi*, an obligatory, gram-negative, intracellular bacterium; the bite of an infected larval chigger containing *O. tsutsugamushi* leads to human infection [1]. Due to its high morbidity and mortality, scrub typhus infection is considered to be serious and life-threatening [2]; the case fatality rate (CFR) can reach as high as 70% if untreated (median being 6%) [3,4]. Scrub typhus mortality is influenced by the virulence of the strains, patient characteristics, delay in diagnosis, and drug resistance [3,5].

Eschar, a characteristic skin lesion and a key diagnostic marker [6] of scrub typhus, is often seen at the site of the chigger bite [7]. Eschar often goes unnoticed, as the vector bite is painless, despite being pruritic on occasion [8]. However, the presence or absence of eschar does not alter the severity of the disease outcome [3]. From the eschar stage, the infection advances to an acute febrile illness [9] with varied clinical presentations such as fever, myalgia, nausea, rash, lymphadenopathy, vomiting, diarrhea, cough, and breathlessness [10]. Mostly in untreated cases, the disease progresses to different organs and organ systems [11,12], and in some cases, is clinically manifested as multi-organ dysfunction syndrome [7].

Serosurveys carried out in scrub typhus endemic regions have also revealed the endemicity of other rickettsial infections caused by the typhus group (TG)—(Epidemic louse-borne typhus and Endemic/ Murine typhus), and spotted fever group (SFG)—(Rocky Mountain spotted fever, Rickettsialpox, and Indian tick typhus) rickettsiae [13–15]; these infections share almost similar signs and symptoms with scrub typhus [15]. Just like scrub typhus, other rickettsial infections are associated with significant mortality unless diagnosed and treated early [16].

In India, after a gap of 65 years, scrub typhus is re-emerging in different regions [17]. Sporadic cases have been reported across the country, even in pockets that had no earlier history of scrub typhus [18,19]. This resurgence may be attributed to changes in climate, agricultural

practices favoring mite breeding, urbanization, migration of infected persons, and increased awareness among clinicians [19].

Mizoram, the landlocked North-East Indian state bordering Myanmar and Bangladesh to its east and west, respectively, is the worst affected [20,21]. With a forest cover of 88.93%, Mizoram is a biodiversity hotspot, and ∼60% of its population is engaged in agricultural practices [22]. In the last five years (2018–2022), Mizoram has seen a substantial increase in scrub typhus incidence; Mizoram's 19,651 cases in the last five years are higher than the rest of India's decadal (2010–2020) cumulative count (18,781) [23]. The emergence or re-emergence of scrub typhus in Mizoram could be associated with the change in vegetation therein [24]. The incidence of bamboo flowering in the state (2007–2009) could have led to an increase in the rodent population, as rodents feed on nutritious fruits and flowers of bamboo [24]. Infected rodents from outside the state might have also migrated due to the availability of food [24]. During the same period, medical practitioners have reported treating several sporadic cases of scrub typhus like-illness [24]. A recent seroprevalence survey reports 46% of the rodents in Mizoram to be positive for scrub typhus antibodies, and 12 scrub typhus outbreaks across Mizoram from 2015–19 [25].

Given the rapid spread of scrub typhus and other rickettsial infections in the state, the Mizoram Government integrated screening and line listing (an organized detailing of cases and associated variables) [26] of scrub typhus and other rickettsial infections from 2018 onwards across all its health centers. The systematic diagnosis and surveillance have helped identify the true burden of these infections, which could have easily gone undetected or misdiagnosed. Here, we report the first comprehensive epidemiological study of scrub typhus and other rickettsial infections from 2018–2022, systematically recorded across a state in India.

## Methods

### Ethics statement

The IRB that provided approval or exemption for the research described was the Directorate of Health Services, Government of Mizoram (NO. D.32020/412014-DHS/IDSP). The reason consent was not obtained was because secondary data analysis was performed using de-identified data.

### Data source

Five years line listing data of scrub typhus and other rickettsial infections from 2018–2022 were obtained from Integrated Disease Surveillance Program (IDSP) under the Health & Family Welfare Department, Government of Mizoram (S1 File). The line listing format includes information on name, age, sex, address, occupation, general symptoms, presence/absence of eschar, date of diagnosis, diagnostic kit utilized, hospitalized or not, and outcome (recovered/ expired). The line listing data were collected from all Government primary health centers (PHCs), community health centers (CHCs), district hospitals (DHs), and private hospitals. Until 2018, screening was mostly done using the scrub typhus-specific Immunochromatographic test (ICT) - InBios rapid test (SD Bioline Tsutsugamushi test, SD Diagnostics, Hagaldong, Kyonggi-do, Korea) [27]. However, as there were widespread reports of scrub typhus-like illness which were testing negative with the ICT across Mizoram, Government of Mizoram introduced and distributed the Weil-Felix (WF) test kits to all its testing units from 2019 onwards; reaction with OXK antigen is suggestive of scrub typhus, whereas reaction with OX19 and OX2 antigens suggests infection by TG and SFG, respectively [28,29]. Even though the commercially available ICTs have a pooled sensitivity and specificity of 66% and 92%, respectively [30], (higher than WF), chances of misdiagnosis can occur with ICTs as they can

detect IgG antibodies in patients who might have had secondary infection [24,31]. Despite its low sensitivity, the WF test serves as a potential tool of diagnosis in resource-limited settings and has proven very useful in guiding clinicians to proceed with appropriate treatment [31]. Based on WF positivity (positive cut-off, 1:160 and above), rickettsial infections were classified into scrub typhus (OXK), other rickettsial infections (OX2, OX19, OX2, and OX19), mixed infections of scrub typhus with other rickettsiae (OXK and OX2, OXK, and OX19, and OXK, OX2, and OX19).

## Statistics

Based on the line listing data of rickettsial infections from 2018 to 2022, a preliminary analysis, including frequencies of the recorded socio-demographic and clinical variables and their association with rickettsial infections (chi-square test) was carried out using the IBM statistical software, SPSS (version 16.0). The chi-square analysis was also carried out to determine the association between eschar and scrub typhus outcome.

A multivariate logistic regression analysis was run to estimate the association of demographic (age, sex, place of residence, occupation) and clinical (eschar and hospitalization) variables with the disease outcome (death or recovery). Age was grouped into 5 categories (<11 years, 11–30 years, 31–50 years, 51–70 years, and 71 and more). Districts were categorized into West (Mamit, Lunglei, Lawngtlai, Kolasib), Central (Aizawl, Serchhip), and East (Champhai, Khawzawl, Saitual, Hnahthial, Siaha). There was no issue of multicollinearity between independent variables. Adjusted odds ratios (aOR) and unadjusted odds ratios (uOR) with a confidence interval (CI) of 95% were calculated; a p-value <0.05 is considered significant. Incidence rates for rickettsial infections were calculated per 1000 persons [32] with a confidence interval (CI) of 95% [33] and were computed by retrieving datasets of the districts' projected population estimates, 2020 [34].

## Spatial mapping and GIS analysis

To understand the spatial distribution of rickettsial infections, the patients' addresses in the line listing data were converted into a spatial database. After the data cleaning process, the addresses of scrub typhus cases were searched in the Google map database for location information. A total of 836 case locations were identified, and their corresponding latitudes and longitudes were extracted (S2 File). The extracted locations were assigned a unique spatial ID and joined with the case details to prepare a spatial database using the ArcGIS 10.4 software (https://desktop.arcgis.com). The total cases recorded in each case location from 2018–2022 were used to identify hotspots of scrub typhus in Mizoram using the Optimized Hotspot Analysis tool in the ArcGIS Pro 3.0 software, as detailed in Supplementary File (S3 File) [35,36]. The hotspots were prepared as dot distribution maps to represent the clusters of rickettsial infections with multiple significant (90%, 95%, and 99% confidence intervals) and non-significant levels. Furthermore, spatial mapping was carried out to illustrate the distribution of incidence rates of rickettsial infections (cases per 1,000 persons per year) across the districts of Mizoram (2018–2022). The district-wise geographic and socio-demographic variables of the study region have been provided in our earlier study [35].

## Results

### Incidence and distribution of rickettsial infections across the different districts of Mizoram

In Mizoram, from 2018–2022, 22,914 cases of rickettsial infections, reactive for either the Tsutsugamushi test or the Weil-Felix test were line-listed from the eleven districts in the state. Among the 22,914 cases, 19,651 were considered to be scrub typhus based on Tsutsugamushi or Weil-Felix OXK positivity. The maximum number of scrub typhus cases were reported from the Aizawl district (9,863 cases, 50.19%), and was followed by Serchhip (2,975 cases, 15.14%), while Champhai (2.07%) and Khawzawl (2.31%) reported the lowest (Table 1 and Fig 1). The cases increased from 2018 to 2019 and were followed by a decline in 2020 and 2021. However, in 2022, there was an upsurge in scrub typhus (6,542 cases), the highest in the past five years (S1 Table and Fig 1). S2 Table details the district-wise distribution of different rickettsial infections diagnosed by the Immunochromatographic test (ICT) and WF test, from 2018 to 2022. Comparatively, more number of scrub typhus cases were diagnosed by ICT than WF test. ICT based diagnosis reported a higher proportion (40.27%) of scrub typhus infections in 2019 (S2 Table). The test results showed a substantial increase in the scrub typhus (OXK positive) caseload; from 297 cases in 2019, it has increased more than 13 times to 3,932 cases in 2022 (S2 Table). A fourfold increase in mixed infections is also seen from 2019 (141 cases) to 2022 (562); by 2022, all the districts have reported mixed infections (S2 Table). The caseload (472) of other rickettsial infections (OX2, OX19, and OX2 and OX19 reactive), though higher than scrub typhus in 2019 (297), were outnumbered by scrub typhus cases in the following years (2020–22) (S2 Table). During the study period, the trend of other rickettsial infections was inconsistent; these are higher in the districts of Champhai, Hnahthial, Khawzawl, Kolasib, Mamit, and Lunglei (S2 Table).

The incidence rates of rickettsial infections (scrub typhus, other rickettsial infections, and mixed infections) are shown in Tables 2, S3, S4 and Fig 2; as scrub typhus constitutes the majority of infections, the incidence rate for scrub typhus and rickettsial infections are similar. The average incidence of rickettsial infections in Mizoram from 2018–2022 is 3.54 cases per 1000 persons-year (95% CI 3.49–3.59) (Table 2 and Fig 2). The average incidence rate of scrub typhus cases in Mizoram is 3.04 cases per 1000 persons-year. Serchhip's average incidence rate (9.34; 95% CI 9.01–9.68) is much higher than the state's average (3.04; 95% CI 2.99–3.08) and is almost double that of Aizawl (4.32; 95% CI 4.23–4.40) (S3 Table). Besides Serchhip and Aizawl, Saitual reported a slightly higher incidence rate (3.12) than the state's average. Among all the districts, Champhai reported the least incidence of scrub typhus (0.79 per 1000 persons-year; 95% CI 0.71–0.86) (S3 Table). The districts of Hnahthial and Khawzawl, have reported higher incidences of other rickettsial infections, with average incidence rates of 2.45 (95% CI 2.19–2.70) and 3.03 (95% CI 2.75–3.31) per 1000 persons-year, respectively (S3 Table). The incidence of mixed infections (scrub typhus and other rickettsial infections) was relatively less during 2019–2021, however, in 2022, a notable rise in the incidence rates was observed across Hnahthial (1.93), Khawzawl (1.45), Serchhip (1.19), and Mamit (1.06) (S3 Table). The incidence rate of OX2 was high in Hnahthial (2.17; 95% CI 1.93–2.42), while for OX19, it was Khawzawl (1.83; 95% CI 1.61–2.05) (S4 Table).

The spatial distribution shows the scrub typhus cases are predominately distributed in the populated (North-Central) districts, especially along the arterial route between Aizawl and Serchhip (Fig 3A). Even though urban centers have higher case locations, loci with >100 cumulative cases are generally distributed in the urban outskirts (Fig 3A). Scrub typhus hotspots in Mizoram are shown in Fig 3B. Most of the settlements in the Saitual district are primary hotspots of scrub typhus (99% significant). The primary hotspots also extended into the

**Table 1. Socio-demographic and clinical profile of rickettsial infections in Mizoram (2018–2022).**

| | Rickettsial infections | | | Total | p-value |
|---|---|---|---|---|---|
| | Scrub typhus* | Other rickettsial infections[†] | Mixed (scrub typhus and other rickettsial) infections[‡] | | |
| | 19651 (85.76%) | 1846 (8.06%) | 1417 (6.18%) | 22914 (100%) | |
| **Age group (in Years)** | n (%) | n (%) | n (%) | n (%) | |
| Less than 11 | 2374 (12.08) | 168 (9.10) | 140 (9.88) | 2682 (11.70) | |
| 11–30 | 4830 (24.58) | 422 (22.86) | 324 (22.87) | 5576 (24.33) | |
| 31–50 | 6731 (34.25) | 673 (36.46) | 484 (34.16) | 7888 (34.42) | 0.000 |
| 51–70 | 4421 (22.50) | 418 (22.64) | 361 (25.48) | 5200 (22.69) | |
| 71 and above | 1295 (6.59) | 165 (8.94) | 108 (7.62) | 1568 (6.84) | |
| **Occupation** | | | | | |
| Farmer | 9805 (49.90) | 1048 (56.77) | 771 (54.41) | 11624 (50.73) | |
| Business | 1955 (9.95) | 109 (5.90) | 124 (8.75) | 2188 (9.55) | |
| Construction workers | 1104 (5.62) | 111 (6.01) | 70 (4.94) | 1285 (5.61) | |
| Government services | 1344 (6.84) | 115 (6.23) | 91 (6.42) | 1550 (6.76) | 0.000 |
| Pre school | 968 (4.93) | 47 (2.55) | 64 (4.52) | 1079 (4.71) | |
| Student | 4301 (21.89) | 402 (21.78) | 279 (19.69) | 4982 (21.74) | |
| Others (aged 71 and above) | 174 (0.89) | 14 (0.76) | 18 (1.27) | 206 (0.90) | |
| **Sex** | | | | | |
| Male | 10278 (52.30) | 890 (48.21) | 721 (50.88) | 11889 (51.89) | 0.003 |
| Female | 9373 (47.70) | 956 (51.79) | 696 (49.12) | 11025 (48.11) | |
| **Districts** | | | | | |
| Aizawl | 9863 (50.19) | 149 (8.07) | 568 (4.008) | 10580 (46.17) | |
| Champhai | 407 (2.07) | 164 (8.88) | 30 (2.12) | 601 (2.62) | |
| Hnahthial | 531 (2.70) | 350 (18.96) | 108 (7.62) | 989 (4.32) | |
| Khawzawl | 453 (2.31) | 452 (24.49) | 76 (5.36) | 981 (4.28) | |
| Kolasib | 459 (2.34) | 191 (10.35) | 97 (6.85) | 747 (3.26) | |
| Lawngtlai | 1357 (6.91) | 10 (0.54) | 196 (13.83) | 1563 (6.82) | 0.000 |
| Lunglei | 1125 (5.72) | 143 (7.75) | 19 (1.34) | 1287 (5.62) | |
| Mamit | 1082 (5.51) | 236 (12.78) | 176 (12.42) | 1494 (6.52) | |
| Saitual | 894 (4.55) | 83 (4.50) | 66 (4.66) | 1043 (4.55) | |
| Serchhip | 2975 (15.14) | 57 (3.09) | 80 (5.65) | 3112 (13.58) | |
| Siaha | 505 (2.57) | 11 (0.60) | 1 (0.07) | 517 (2.26) | |
| **Hospitalization** | | | | | |
| Yes | 3832 (19.50) | 148 (8.02) | 202 (14.26) | 4182 (18.25) | 0.000 |
| No | 15819 (80.50) | 1698 (91.98) | 1215 (85.74) | 18732 (81.75) | |
| **Presence of eschar** | | | | | |
| Yes | 443 (2.25) | 92 (4.98) | 46 (3.25) | 581 (2.54) | 0.000 |
| No | 19208 (97.75) | 1754 (95.02) | 1371 (96.75) | 22333 (97.46) | |

p<0.05 is significant

*Scrub typhus: Rapid tsutsugamushi and/or WF OXK positive

[†]Other rickettsial infections: WF OX2 and/or OX19 positive

[‡]Mixed infections: WF OXK with OX2 and/or OX19 positive

Aizawl district, especially along the borders of the Saitual and Aizawl districts. The capital city of Mizoram, Aizawl, and its neighboring major town, Serchhip are secondary hotspots for rickettsial infections. Siaha, Lawngtlai, and Lunglei districts are coldspots of scrub typhus; these locations have less than 25 total cases during 2018–2022.

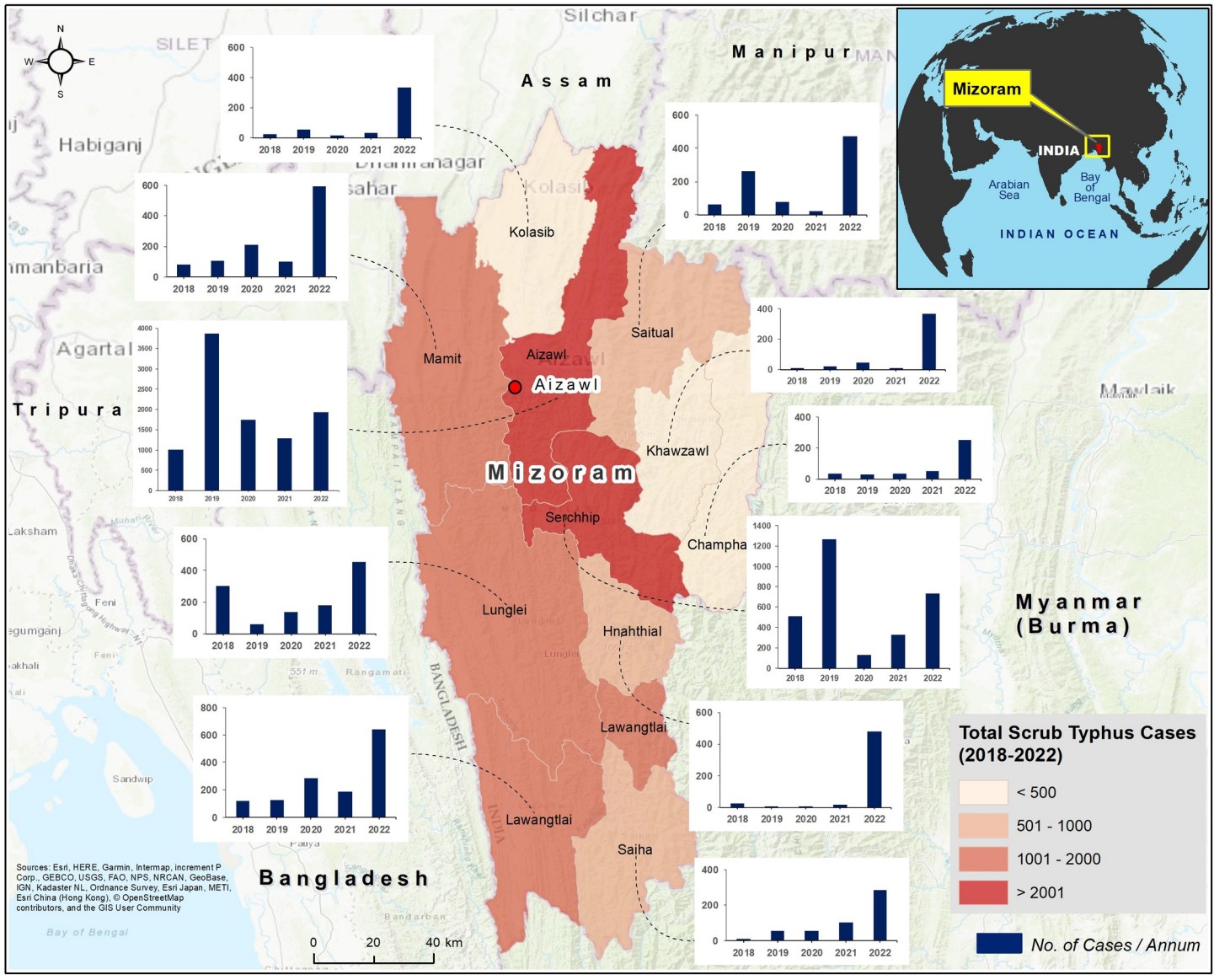

**Fig 1. Distribution of scrub typhus cases across the districts of Mizoram (2018–2022).** The trends in case distribution are represented as located bar chart. The y-axis of the graph shows the total cases in each year from 2018–2022. The length of the y-axis varies according to the caseloads across the districts of Mizoram. The background base map represents topography. For more information about the base map services, visit http://goto.arcgisonline.com/maps/World_Topo_Map.

## Seasonal distribution of scrub typhus cases in Mizoram

A spike in scrub typhus cases was observed during the monsoon and post-monsoon seasons (June to November) (Fig 4). Together, monsoon and post-monsoon months account for more than 60% of the cases. July accounted for the maximum number of cases (2,641, 13.44%), followed by August (2,494, 12.7%). Aside from monsoon and post-monsoon months, scrub typhus cases were reported more in January (1,794, 9.13%). Cases were also reported during the summer (March to May), accounting for 17.12% of the total cases (Fig 4 and S1 Table).

**Table 2. Incidence rates of rickettsial infections across the districts of Mizoram (2018–2022).**

| Cases per 1000 persons-year | Aizawl | Champhai | Hnahthial | Khawzawl | Kolasib | Lawngtlai | Lunglei | Mamit | Saitual | Serchhip | Siaha | Mizoram |
|---|---|---|---|---|---|---|---|---|---|---|---|---|
| Rickettsial infections* | | | | | | | | | | | | |
| 2018 | 2.21 (1010, 2.07–2.35) | 0.33 (34, 0.22–0.44) | 0.76 (27, 0.47–1.04) | 0.24 (9, 0.08–0.40) | 0.21 (23, 0.13–0.30) | 0.80 (119, 0.66–0.95) | 2.17 (300, 1.92–2.41) | 0.80 (78, 0.62–0.97) | 1.08 (62, 0.81–1.35) | 8.05 (513, 7.36–8.75) | 0.17 (8, 0.05–0.29) | 1.69 (2183, 1.62–1.76) |
| 2019 | 8.69 (3974, 8.42–8.96) | 0.96 (99, 0.77–1.14) | 3.55 (127, 2.93–4.17) | 7.38 (275, 6.51–8.25) | 0.89 (97, 0.72–1.07) | 0.95 (140, 0.79–1.10) | 0.43 (59, 0.32–0.54) | 1.20 (118, 0.99–1.42) | 4.53 (259, 3.98–5.08) | 19.88 (1266, 18.78–20.97) | 1.23 (58, 0.91–1.55) | 5.00 (6472, 4.88–5.12) |
| 2020 | 4.30 (1965, 4.11–4.49) | 0.80 (83, 0.63–0.97) | 4.36 (156, 3.68–5.05) | 5.45 (203, 4.70–6.20) | 1.05 (114, 0.86–1.24) | 2.26 (334, 2.02–2.50) | 1.50 (207, 1.29–1.70) | 3.90 (382, 3.51–4.29) | 1.38 (79, 1.08–1.69) | 2.10 (134, 1.75–2.46) | 1.27 (60, 0.95–1.59) | 2.87 (3717, 2.78–2.96) |
| 2021 | 3.39 (1548, 3.22–3.56) | 0.99 (103, 0.80–1.19) | 2.32 (83, 1.82–2.82) | 0.70 (26, 0.43–0.97) | 1.19 (129, 0.98–1.39) | 1.93 (286, 1.71–2.16) | 1.70 (235, 1.48–1.92) | 1.66 (163, 1.41–1.92) | 1.61 (92, 1.28–1.94) | 5.51 (351, 4.93–6.09) | 2.20 (104, 1.78–2.63) | 2.41 (3120, 2.33–2.49) |
| 2022 | 4.56 (2083, 4.36–4.75) | 2.72 (282, 2.41–3.04) | 16.67 (596, 15.33–18.01) | 12.56 (468, 11.42–13.70) | 3.54 (384, 3.18–3.89) | 4.62 (684, 4.28–4.97) | 3.51 (486, 3.20–3.83) | 7.69 (753,7.14–8.23) | 9.63 (551, 8.83–10.43) | 13.31 (848, 12.42–14.21) | 6.08 (287, 5.38–6.79) | 5.73 (7422, 5.60–5.86) |
| Avg. incidence rate | 4.63 (10580, 4.54–4.72) | 1.16 (601, 1.07–1.25) | 5.53 (989, 5.19–5.88) | 5.27 (981, 4.94–5.60) | 1.38 (747, 1.28–1.47) | 2.11 (1563, 2.01–2.22) | 1.86 (1287, 1.76–1.96) | 3.05 (1494, 2.90–3.20) | 3.65 (1043, 3.42–3.87) | 9.77 (3112, 9.43–10.12) | 2.19 (517, 2.00–2.38) | 3.54 (22914, 3.49–3.59) |

*Positive by rapid tsutsugamushi and/or WF (OXK/OX2/OX19) test

## Socio-demographic, occupational and clinical profile of rickettsial infections across the different districts of Mizoram

All age groups (from less than one year to greater than 91 years) were affected; the highly affected group being 31 to 50 years (34.42%), followed by 11 to 30 years (24.33%); the youngest was one month and oldest was 98 years (S5 Table and S1 Fig). The proportion of reported cases was slightly higher in males (51.89%) when compared to females (48.11%) (Tables 1 and S5). Males were higher than females in scrub typhus and mixed infections, while female cases were more in other rickettsial infections (Tables 1 and S5). When stratified based on occupation, 50.73% and 21.74% of the affected individuals were farmers and students, respectively (S6 Table). Individuals in the fields of business, construction work, and government service make up 21.9% of the reported cases (S6 Table).

The most common clinical presentations were fever/persistent fever (85.03%), headache (29.35%), rash (26.80%), chills (21.36%), cough (9.21%), body ache (8.58%), and nausea (3.23%) (S7 Table). Clinical presentations pertaining to CNS (central nervous system) involvement were seen in ∼2% of the cases; the proportion was higher in OX19-positive cases (S7 Table). Eschar, the characteristic lesion, was found in 443 scrub typhus cases, 92 other rickettsial infections, and 46 mixed infection cases (Table 1). While just ∼2% of the reported scrub typhus cases had eschar as a clinical presentation, >7% of the OX2 reactive cases reported eschar (S7 Table). Despite the nature of the occupation, the distribution of clinical presentations among the different classes was the same (S8 Table).

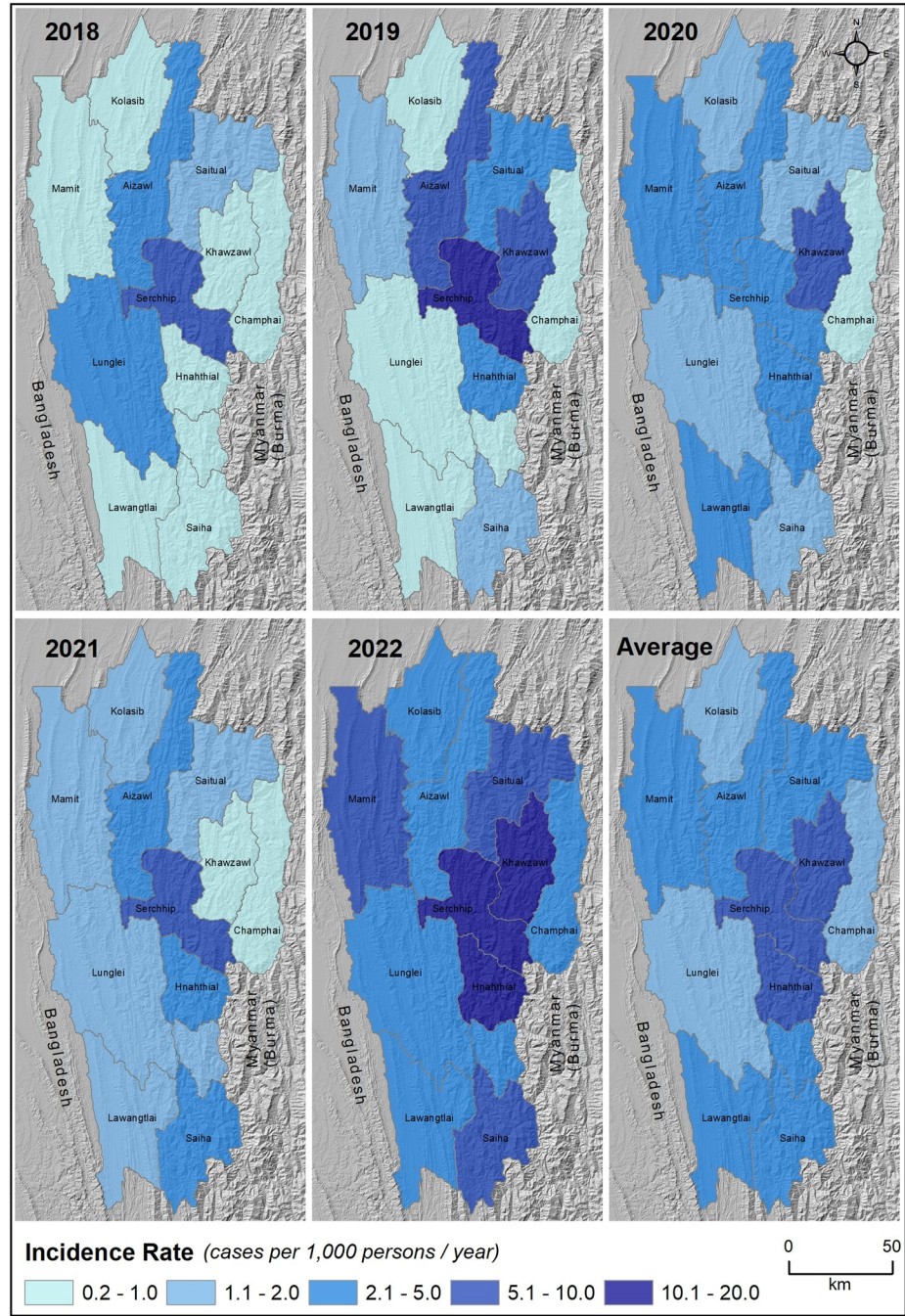

**Fig 2. Spatial distribution of incidence rates of rickettsial infections across the districts of Mizoram (2018–2022).**
Year-wise (2018 to 2022) and average incidence rates are represented spatially as cases per 1,000 persons per year. The darker (blue) shades in the central districts of Hnahthial, Serchhip, and Khawzawl indicate a very high incidence rate (>10 cases per 1,000 persons per year). Serchhip district showed the highest incidence rate throughout the period (except the year 2020). The background base map represents the topography of Mizoram and surroundings. The topographic layer was prepared using NASA SRTM (Shuttle Radar Topography Mission) Global 30 arc second DEM data accessed from https://doi.org/10.5067/MEaSUREs/SRTM/SRTMGL30.002. The map layout was created using a licensed version of ArcGIS 10.4 software by Esri (https://desktop.arcgis.com).

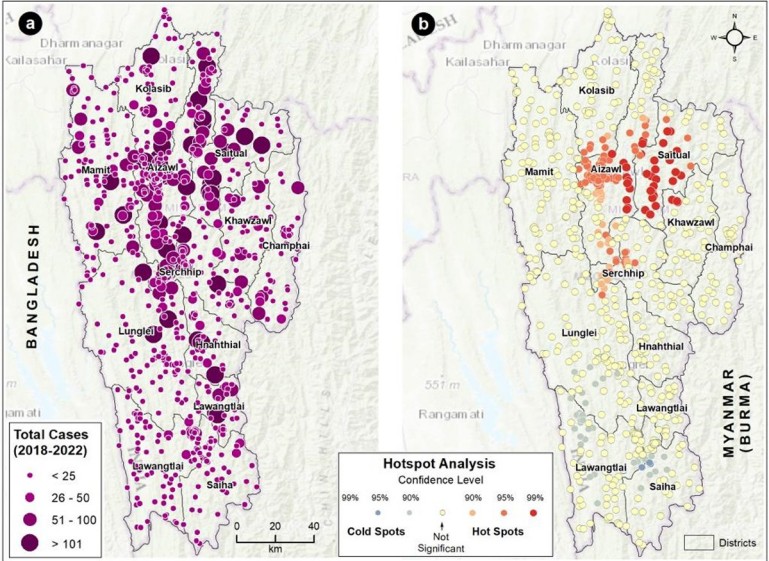

**Fig 3. Distribution of (a) scrub typhus cases (2018–2022) and (b) hotspots in Mizoram.** The large dot with dark purple color (a) indicates higher caseloads (>101). The color (purple) intensity and size of the dot gradually decrease with a reduction in the caseload. The red and blue shaded dots represent (b) the hot and cold spots, respectively, of scrub typhus cases in Mizoram with different statistical significance. The yellow-colored dots are not statistically significant for forming hot or cold spots. The background base map represents topography. For more information about the base map services, visit http://goto.arcgisonline.com/maps/World_Topo_Map.

Among the reported scrub typhus cases, 19.5% of individuals were hospitalized; the distribution was the same across males and females (S9 Table). Individuals in the age group 31–50 years contributed to >30% of the hospitalized cases (S9 Table). Even though farmers were high among those who sought care, only <20% of them were hospitalized. On the other hand, the highest proportion of hospitalization (63.79%) was observed in working individuals aged 71 and above (S9 Table). Table 3 describes the odds of death due to rickettsial infections and S10 Table details the case fatality rates. Children aged 10 and below have higher odds of death due to rickettsial infections (aOR = 5.441, p<0.05) (Table 3). Also, in this age group, CFR due to scrub typhus is high with ∼5 deaths per 1000 (0.51) and is next only to the age group 51–70 whose CFR is 0.66 (S10 Table). The proportion of rickettsial deaths is more in men (62.96%) than in women (37.04%) (S10 Table). When compared with the Eastern districts of Mizoram, Western, and Central districts have higher odds (aOR = 3.26 and aOR = 2.39, respectively, p<0.05) of rickettsial deaths (Table 3). Among the occupational groups, farmers, businessmen, and construction workers have higher odds (aOR>1, p<0.05) of death if infected with rickettsiae (Table 3). The odds of death are 17 times higher (p<0.001) in hospitalized cases; of the 81 rickettsial related deaths occurred, 76.54% (n = 62) were hospitalized cases (Tables 3 and S10). In the study period, 75 scrub typhus-related deaths were reported; 74.66% of them were hospitalized cases (S9 and S10 Tables). Scrub typhus deaths were significantly associated with eschar (p<0.05) (S11 Table). Also, in rickettsial cases with eschar, the odds of death are 2.5 times higher (p<0.05) (Table 3). During the study period, scrub typhus CFR was 0.38, while it was 0.11 and 0.28, respectively, for other rickettsial and mixed infections (S10 Table).

## Discussion

Despite its importance and public health threat, scrub typhus has stayed under the radar in many parts of India. Studies in the last decade have reported 18,781 confirmed cases of scrub

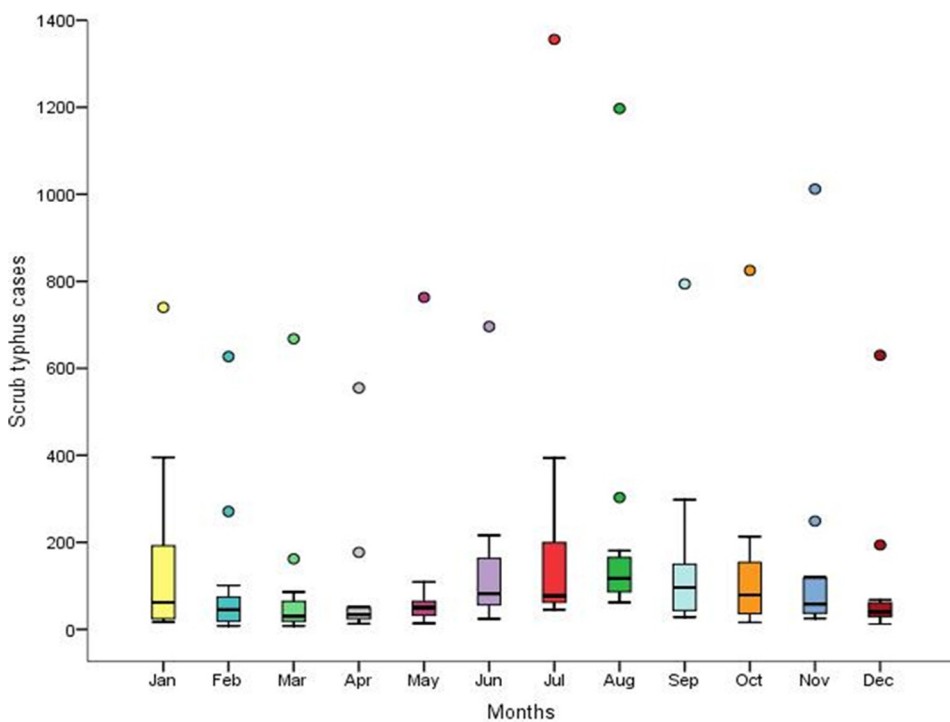

**Fig 4. Monthly distribution of scrub typhus cases across the 11 districts of Mizoram (2018–2022).** The rectangular box represents the interquartile range; bottom line represents first quartile (25th percentile) and the top line represents third quartile (75th percentile). Median is represented by the horizontal line within the box. Extending from the box are the whiskers (vertical lines). The bottom whisker end represents minimum number of cases and top whisker end represents maximum number of cases which isn't an outlier. Outliers (mild and extreme) are represented by dots.

typhus across India; most of the studies are hospital-based (138), and only two are community-based [23]. As it is an acute febrile disease that could be easily treated by commonly used antibiotics, many a time the disease goes undiagnosed. The decadal count of 18,781 cases might just be a fraction of the true prevalence of scrub typhus across India [23]. The abundance of forest cover, rodents' easy access to households, farming activities, and conducive environmental conditions provide the ideal setting for the transmission of rickettsial infections in Mizoram [25]. In Mizoram, the first cases of scrub typhus were officially recorded in 2012 [24]. From 2018 onwards, there was a substantial increase in scrub typhus cases. In this study, the Mizoram Government's initiative to systematically diagnose and record the incidence of scrub typhus and other rickettsial infections across all the testing units has shed important insights on the magnitude of the disease burden in Mizoram; spatial analysis indicates scrub typhus cases are widespread in both urban and rural settings.

Even though there are sporadic reports of scrub typhus across India, there were no systematic recording and reporting at the state level. To our knowledge, this is the first report that details the burden of scrub typhus and other rickettsial infections across an entire state in India. The low rickettsial death rates in Mizoram could be attributed to public awareness and the set-up in place for rapid diagnosis and treatment—key initiatives of the concerned health departments. In India, in the last decade, the case-fatality rate (CFR) of scrub typhus was 6.3%, and in those with multi-organ dysfunction syndrome, the CFR was 38.9% [23]. Delayed presentation, diagnosis, and treatment might be the reasons for the high CFR seen in other settings. With the availability of rapid immunochromatographic diagnostic tests [37,38] (RDT), screening of acute undifferentiated febrile illness (AUFI) for scrub typhus in scrub typhus

**Table 3. Socio-demographic and clinical characteristics associated with rickettsial deaths in Mizoram (2018–22).**

| Variables | | Unadjusted odds ratios (uOR) | | Adjusted odds ratios (aOR) | | | |
|---|---|---|---|---|---|---|---|
| | | Exp (β) | p-value* | Exp (β) | 95% Confidence Interval | | p-value* |
| | | | | | Lower Bound | Upper Bound | |
| **Eschar presence** | | | | | | | |
| | Yes | 2.542 | **0.044** | 2.597 | 1.023 | 6.595 | **0.045** |
| | No (Ref) | 1 | | 1 | | | |
| **Hospitalisation** | | | | | | | |
| | Yes | 14.821 | **0.000** | 16.995 | 10.025 | 28.811 | **0.000** |
| | No (Ref) | 1 | | 1 | | | |
| **Age (in years)** | | | | | | | |
| | Less than 11 | 2.08 | 0.052 | 5.44 | 1.69 | 17.56 | **0.005** |
| | 11–30 (Ref) | 1 | | 1 | | | |
| | 31–50 | 0.91 | 0.788 | 0.50 | 0.23 | 1.11 | 0.089 |
| | 51–70 | 2.31 | **0.010** | 1.26 | 0.60 | 2.66 | 0.538 |
| | 71 and more | 1.27 | 0.646 | 0.68 | 0.20 | 2.26 | 0.528 |
| **Occupation** | | | | | | | |
| | Farmer | 1.072 | 0.827 | 3.984 | 1.179 | 13.464 | **0.026** |
| | Business | 1.957 | 0.089 | 6.693 | 1.898 | 23.605 | **0.003** |
| | Construction workers | 3.627 | **0.001** | 17.949 | 5.081 | 63.407 | **0.000** |
| | Government services | .458 | 0.303 | 1.621 | 0.274 | 9.579 | 0.594 |
| | Pre school | 1.320 | 0.625 | 0.596 | 0.185 | 1.920 | 0.386 |
| | Others (aged 71 and above) | 1.731 | 0.597 | 2.636 | 0.214 | 32.478 | 0.449 |
| | Student (Ref) | 1 | | 1 | | | |
| **Sex** | | | | | | | |
| | Male | 1.579 | **0.047** | 1.559 | 0.981 | 2.480 | 0.060 |
| | Female (Ref) | 1 | | 1 | | | |
| **Districts** | | | | | | | |
| | West | 2.575 | **0.044** | 3.265 | 1.287 | 8.281 | **0.013** |
| | Central | 2.823 | **0.016** | 2.395 | 1.018 | 5.637 | **0.045** |
| | East (Ref) | 1 | | 1 | | | |

*p<0.05 is significant

endemic regions might be rewarding. The sensitivity and specificity of RDT are comparable to ELISA in detecting scrub typhus-specific IgM antibodies [39]. Studies have shown 25.3% of AUFI cases are due to scrub typhus [23]. As carried out for dengue [40] and chikungunya [41], a nationwide serosurvey for scrub typhus could select the endemic regions which could be prioritized for integrating scrub typhus screening at district-level primary health centers. However, as the rapid diagnostic tests are specific for scrub typhus, they will not detect other rickettsial infections. For this reason, in 2019, Mizoram Govt. introduced the WF test across all its testing centers, as patients were reporting scrub typhus-like symptoms, but were negative when tested with the RDT. Even though the WF test has poor sensitivity, it can detect recent or previous infections with the typhus group (TG), scrub typhus group (STG), and spotted fever group (SFG) [28]. The WF screening has identified a substantial number of other rickettsial and mixed infections in Mizoram. This is in line with serosurveillance data from rodents collected across Mizoram; rodents were seropositive for OX2, OXK, and OX19 antigens [25]. Again, we believe this is the first state-wide systematic recording of other rickettsial infections

in India. There were no differences in the clinical presentations between OXK-positive scrub typhus cases and OX2 and/or OX19-positive other rickettsial infections. This indistinguishability calls for increased awareness and the necessitation of rickettsia-specific serosurveys to reduce misdiagnosis and understand the true prevalence in the community. Eschar, though described as the "cutaneous hallmark" [42] is not always present in all the infected individuals [3]; thus lowering the suspicion index (a key to early diagnosis) [15,43]. The prevalence of eschar ranges from 7–80% [9] and is low in South East Asia and India [7]. Dark skin, strain characteristics, and atypical appearances in damp and most skin areas affect the detection and prevalence of eschar [9]. In this study, only 2.5% of the rickettsial cases had eschar, underscoring the unreliability of eschar as a clinical diagnostic marker. The significant association between eschar and death indicates the importance of genome sequencing to correlate strains with clinical severity.

The widespread distribution of rickettsial infections and their continuous antibiotic treatment poses a serious threat of drug resistance in Mizoram. However, because of integrating the screening of rickettsial diseases as part of routine diagnostics across the state, the clinicians might be better placed to detect any deviation in the treatment and recovery period against a specific antibiotic and could course correct. Profiling the genome of the circulating strains and correlating it with disease severity are critical in devising prevention, control, and treatment strategies. As the screening of scrub typhus is not routinely carried out in many parts of the country, indiscriminate antibiotic use against AUFI might lead to the emergence of drug-resistant rickettsiae.

In this study, the majority of the rickettsial infections were reported in monsoon and post-monsoon seasons, and this is in line with several studies carried out in India [44–46]. Similarly, as reported earlier, the majority of the cases were in individuals involved in farming activities [46–48]. In Mizoram, Jhum or shifting cultivation, also known as slash and burn agriculture is the major farming activity, and an estimated 54% are practicing it [49]. Even though 50.73% of the cases are farmers, only 16.33% of them were hospitalized. The reason for low hospitalization among farmers might be partial immunity due to repeated exposure to natural infection in the fields. Natural *Orientia* infection does not elicit sterilizing and long-lasting immunity and does not confer cross-protection against infection from other strains [50]. High hospitalization (62.62%) in working adults >71 years may be because of their lack of prior exposure (the first scrub typhus cases were recorded only in 2012), and age-related immunosenescence [51]. A study in South Korea showed an increase in health expenditure and disease burden to be associated with elderly patients (>65 years), as both complications and mortality were more common in elderly scrub typhus compared to younger ones [52].

Due to the resource-limited settings, the study has several limitations. As the sensitivity of the WF test is low, the true burden of rickettsial infections might be substantially higher. One alternative is to use highly sensitive molecular assays; however, its feasibility in resource-limited settings such as this is to be ascertained. Also, the mixed infections categorized based on the WF test had to be verified by molecular tools. The decline in the caseloads in 2020 and 2021 could be due to the pandemic, as all testing centers were engaged with the diagnosis of Covid-19. Another limitation is the lack of clinical information such as the severity of the disease, length of recovery, and criterion for hospital admission; this information might be important in disease management and could help predict the disease outcome.

Overall, this study details the importance of integrating the screening of rickettsial diseases at the district level across all the health centers. This approach could serve as a template for other states/regions which are endemic to scrub typhus and other rickettsial infections.

## Supporting information

**S1 File. Line-listing of scrub typhus and other rickettsial infections in Mizoram (2018–22).**
(XLSX)

**S2 File. Case locations of rickettsial infections across Mizoram (2018–22).**
(XLSX)

**S3 File. Optimized Hotspot Analysis.**
(DOCX)

**S1 Table. Monthly distribution of scrub typhus cases across the districts of Mizoram (2018–2022).**
(DOCX)

**S2 Table. Diagnostic tests based distribution of rickettsial infections across the districts of Mizoram (2018–22).**
(DOCX)

**S3 Table. Incidence rates of scrub typhus, other rickettsial and mixed infections across the districts of Mizoram (2018–2022).**
(DOCX)

**S4 Table. Incidence rates of rickettsial infections across districts of Mizoram based on antigenic strains (2019–2022).**
(DOCX)

**S5 Table. Age and gender based distribution of rickettsial infections in Mizoram (2018–2022).**
(DOCX)

**S6 Table. Distribution of rickettsial cases across different occupational groups in Mizoram (2018–2022).**
(DOCX)

**S7 Table. Clinical presentations of the reported rickettsial infections in Mizoram (2018–22).**
(DOCX)

**S8 Table. Distribution of clinical presentations among different occupational groups in Mizoram.**
(DOCX)

**S9 Table. Distribution of hospitalized rickettsial and scrub typhus cases in Mizoram (2018–2022).**
(DOCX)

**S10 Table. Case fatality rates (CFRs) of different rickettsial infections in the state of Mizoram, 2018–2022.**
(DOCX)

**S11 Table. Association between eschar and scrub typhus disease outcome.**
(DOCX)

**S1 Fig. Gender based distribution of rickettsial infections across different age groups in Mizoram (2018–22).**
(TIFF)

## Acknowledgments

The authors would like to convey their sincere gratitude to health officials and other concerned Govt. officials of Mizoram for making provision to distribute WF and ICT kits to diagnose scrub typhus and other rickettsial diseases to all Govt. hospitals in the state free of cost. The authors also sincerely thank the Director of Health Services, Govt. of Mizoram, for giving permission to collect and analyze the secondary data. The team would also like to convey their sincere thanks to all district health officials, data managers, and data entry operators under IDSP for their continuous and untiring support in collection, entry, and reporting of the data. The authors also acknowledge the Institutional Advanced-level Biotech Hub (BT/NER/143/ SP44393/2021, dated 18[th] November 2022) Dept. of Zoology, Pachhunga University College, for providing computational and laboratory facilities.

## Author Contributions

**Conceptualization:** Vanramliana, Lalfakzuala Pautu, Praveen Balabaskaran Nina.

**Formal analysis:** Naveen Kumar Kodali, Christiana Amarthaluri, Karuppusamy Balasubramani.

**Writing – original draft:** Vanramliana, Lalfakzuala Pautu, Praveen Balabaskaran Nina.

**Writing – review & editing:** Pachuau Lalmalsawma, Gabriel Rosangkima, Devojit Kumar Sarma, Hunropuia Chinzah, Yogesh Malvi.

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
