## [Decision Letter · Decision Letter 0]

11 Aug 2023

Dear Mr Pautu,

Thank you very much for submitting your manuscript "Epidemiology of scrub typhus and other rickettsial infections (2018-22) in the hyper-endemic setting of Mizoram, North-East India" for consideration at PLOS Neglected Tropical Diseases. As with all papers reviewed by the journal, your manuscript was reviewed by members of the editorial board and by several independent reviewers. In light of the reviews (below this email), we would like to invite the resubmission of a significantly-revised version that takes into account the reviewers' comments. 

We cannot make any decision about publication until we have seen the revised manuscript and your response to the reviewers' comments. Your revised manuscript is also likely to be sent to reviewers for further evaluation.

Sincerely,

Wen-Ping Guo

Academic Editor

Dileepa Ediriweera

Section Editor

Reviewer's Responses to Questions

**Key Review Criteria Required for Acceptance?**

**Methods**

-Are the objectives of the study clearly articulated with a clear testable hypothesis stated?

-Is the study design appropriate to address the stated objectives?

-Is the population clearly described and appropriate for the hypothesis being tested?

-Is the sample size sufficient to ensure adequate power to address the hypothesis being tested?

-Were correct statistical analysis used to support conclusions?

-Are there concerns about ethical or regulatory requirements being met?

Reviewer #1: Yes

Reviewer #2: -Are the objectives of the study clearly articulated with a clear testable hypothesis stated? Yes

-Is the study design appropriate to address the stated objectives? No, The prevalence of each region (county/city) is affected by the population ratio and number of patients. Spatial mapping and GIS analysis are preferably described by incidence rates. 

-Is the population clearly described and appropriate for the hypothesis being tested? No. Same answer as previous question

-Is the sample size sufficient to ensure adequate power to address the hypothesis being tested? Yes

-Were correct statistical analysis used to support conclusions? Yes

-Are there concerns about ethical or regulatory requirements being met? No

**Results**

-Does the analysis presented match the analysis plan?

-Are the results clearly and completely presented?

-Are the figures (Tables, Images) of sufficient quality for clarity?

Reviewer #1: Yes

Reviewer #2: -Does the analysis presented match the analysis plan? Yes

-Are the results clearly and completely presented? No, In table 3, the adjusted odds ratio of death in Districts are preferably analyzed by case fatal rate calculated per 1000 persons. 

-Are the figures (Tables, Images) of sufficient quality for clarity? No, they need to be revised as comments above.

**Conclusions**

-Are the conclusions supported by the data presented?

-Are the limitations of analysis clearly described?

-Do the authors discuss how these data can be helpful to advance our understanding of the topic under study?

-Is public health relevance addressed?

Reviewer #1: Yes.

Reviewer #2: -Are the conclusions supported by the data presented? Yes

-Are the limitations of analysis clearly described? Yes

-Do the authors discuss how these data can be helpful to advance our understanding of the topic under study? Yes

-Is public health relevance addressed? Yes

**Editorial and Data Presentation Modifications?**

Reviewer #1: -

Reviewer #2: -It is better to indicate references in the sentences on page 5, lines 106 to 108.

-In methods, The positive cut-off of the Weil-Felix test should be described.

-In Table 1, it would be useful to describe the differences in the diagnosis test of scrub typhus between 2018 and after 2018. In particular, in 2018, it would be better to describe only the diagnosis of scrub typhus.

- The spatial distribution information will be of more value if the terrain in those provinces is explained.

**Summary and General Comments**

Reviewer #1: The research paper by Vanramliana et al. (PNTD-D-23-00849, Epidemiology of scrub typhus and other rickettsial infections (2018-22) in the hyperendemic setting of Mizoram, North-East India) provides the results of screening and line listing of scrub typhus and related infections in Mizoram. They say that low fatality of 0.35 as against ∼6% from other locations could be used as a template for disease monitoring and management.

Specific points:

“line-listed data positive for” – the authors should provide information on the total number of different screening tests done, year-wise and region-wise. That will provide the % of positive cases.

“836 case locations were identified” – it is unclear if authors found coordinates for only 836 cases or 836 distinct coordinates were found. How many cases could not be mapped? They may include coordinate information in S1 file.

Table 1 p-value – for which column or row and what test?

Table 2 – how incident rate range was calculated? Give year-wise population size for those places in supplemental.

Is there data/information on severity of the disease and length to recovery? Also, what decides hospitalization after screening? These data might be important for disease management and might influence the outcome/death. No such info is given in the S1 file.

“76.54% of the hospitalized rickettsial cases suffered death” – seems an erroneous statement. That will be 3000+ deaths! Check.

Reviewer #2: The strengths of the article lie in the structure and epidemiological descriptions. However, the data analysis (esp, Spatial Mapping and GIS analysis) is incomplete. The map should provide incidence rates instead of total cases in each regions.

PLOS authors have the option to publish the peer review history of their article (what does this mean?). If published, this will include your full peer review and any attached files.

Reviewer #1: Yes: R. Shyama Prasad Rao

Reviewer #2: No

Figure Files:

Data Requirements:

Please note that, as a condition of publication, PLOS' data policy requires that you make available all data used to draw the conclusions outlined in your manuscript. Data must be deposited in an appropriate repository, included within the body of the manuscript, or uploaded as supporting information. This includes all numerical values that were used to generate graphs, histograms etc.. For an example see here: http://www.plosbiology.org/article/info:doi%2F10.1371%2Fjournal.pbio.1001908#s5.
---

## [Decision Letter · Decision Letter 1]

18 Sep 2023

Dear Mr Pautu,

Thank you very much for submitting your manuscript "Epidemiology of scrub typhus and other rickettsial infections (2018-22) in the hyper-endemic setting of Mizoram, North-East India" for consideration at PLOS Neglected Tropical Diseases. As with all papers reviewed by the journal, your manuscript was reviewed by members of the editorial board and by several independent reviewers. In light of the reviews (below this email), we would like to invite the resubmission of a significantly-revised version that takes into account the reviewers' comments. 

Still the authors have to address reviewer 1 queries. Please address comprehensively to the queries, particularly how you assess and incorportate spatial correlation into the models.

We cannot make any decision about publication until we have seen the revised manuscript and your response to the reviewers' comments. Your revised manuscript is also likely to be sent to reviewers for further evaluation.

Sincerely,

Dileepa Ediriweera

Section Editor

Dileepa Ediriweera

Section Editor

Still the authors have to address reviewer 1 queries. Please address comprehensively to the queries, particularly how you assess and incorportate spatial correlation into the models.

Reviewer's Responses to Questions

**Key Review Criteria Required for Acceptance?**

**Methods**

-Are the objectives of the study clearly articulated with a clear testable hypothesis stated?

-Is the study design appropriate to address the stated objectives?

-Is the population clearly described and appropriate for the hypothesis being tested?

-Is the sample size sufficient to ensure adequate power to address the hypothesis being tested?

-Were correct statistical analysis used to support conclusions?

-Are there concerns about ethical or regulatory requirements being met?

Reviewer #1: -

Reviewer #2: Yes. the revised manuscript has been revised according to reviewer's comments.

**Results**

-Does the analysis presented match the analysis plan?

-Are the results clearly and completely presented?

-Are the figures (Tables, Images) of sufficient quality for clarity?

Reviewer #1: -

Reviewer #2: Yes. the revised manuscript has been revised according to reviewer's comments.

**Conclusions**

-Are the conclusions supported by the data presented?

-Are the limitations of analysis clearly described?

-Do the authors discuss how these data can be helpful to advance our understanding of the topic under study?

-Is public health relevance addressed?

Reviewer #1: -

Reviewer #2: Yes

**Editorial and Data Presentation Modifications?**

Reviewer #1: -

Reviewer #2: Yes. the revised manuscript has been revised according to reviewer's comments.

**Summary and General Comments**

Reviewer #1: The authors have responded to the reviewers’ comments and revised the manuscript accordingly.

Reviewer #2: The strengths of the article lie in the structure and epidemiological descriptions. The data analysis (esp, Spatial Mapping and GIS analysis) is improved in revised manuscript. The map including incidence rates in each regions is more imformative in revised manuscript.

PLOS authors have the option to publish the peer review history of their article (what does this mean?). If published, this will include your full peer review and any attached files.

Reviewer #1: Yes: R. Shyama Prasad Rao

Reviewer #2: No

Figure Files:

Data Requirements:

Please note that, as a condition of publication, PLOS' data policy requires that you make available all data used to draw the conclusions outlined in your manuscript. Data must be deposited in an appropriate repository, included within the body of the manuscript, or uploaded as supporting information. This includes all numerical values that were used to generate graphs, histograms etc.. For an example see here: http://www.plosbiology.org/article/info:doi%2F10.1371%2Fjournal.pbio.1001908#s5.
---

## [Editor Report · Decision Letter 2]

27 Sep 2023

Dear Mr Pautu,

We are pleased to inform you that your manuscript 'Epidemiology of scrub typhus and other rickettsial infections (2018-22) in the hyper-endemic setting of Mizoram, North-East India' has been provisionally accepted for publication in PLOS Neglected Tropical Diseases.

Best regards,

Wen-Ping Guo

Academic Editor

Dileepa Ediriweera

Section Editor

---

## [Editor Report · Acceptance letter]

17 Oct 2023

Dear Mr Pautu,

We are delighted to inform you that your manuscript, "Epidemiology of scrub typhus and other rickettsial infections (2018-22) in the hyper-endemic setting of Mizoram, North-East India," has been formally accepted for publication in PLOS Neglected Tropical Diseases.

Best regards,

Shaden Kamhawi

co-Editor-in-Chief

Paul Brindley

co-Editor-in-Chief
